# Preliminary Identification of Potential Vaccine Targets for the COVID-19 Coronavirus (SARS-CoV-2) Based on SARS-CoV Immunological Studies

**DOI:** 10.3390/v12030254

**Published:** 2020-02-25

**Authors:** Syed Faraz Ahmed, Ahmed A. Quadeer, Matthew R. McKay

**Affiliations:** 1Department of Electronic and Computer Engineering, The Hong Kong University of Science and Technology, Hong Kong, China; sfahmed@connect.ust.hk; 2Department of Chemical and Biological Engineering, The Hong Kong University of Science and Technology, Hong Kong, China

**Keywords:** Coronavirus, 2019-nCoV, 2019 novel coronavirus, SARS-CoV-2, COVID-19, SARS-CoV, MERS-CoV, T cell epitopes, B cell epitopes, vaccine

## Abstract

The beginning of 2020 has seen the emergence of COVID-19 outbreak caused by a novel coronavirus, Severe Acute Respiratory Syndrome Coronavirus 2 (SARS-CoV-2). There is an imminent need to better understand this new virus and to develop ways to control its spread. In this study, we sought to gain insights for vaccine design against SARS-CoV-2 by considering the high genetic similarity between SARS-CoV-2 and SARS-CoV, which caused the outbreak in 2003, and leveraging existing immunological studies of SARS-CoV. By screening the experimentally-determined SARS-CoV-derived B cell and T cell epitopes in the immunogenic structural proteins of SARS-CoV, we identified a set of B cell and T cell epitopes derived from the spike (S) and nucleocapsid (N) proteins that map identically to SARS-CoV-2 proteins. As no mutation has been observed in these identified epitopes among the 120 available SARS-CoV-2 sequences (as of 21 February 2020), immune targeting of these epitopes may potentially offer protection against this novel virus. For the T cell epitopes, we performed a population coverage analysis of the associated MHC alleles and proposed a set of epitopes that is estimated to provide broad coverage globally, as well as in China. Our findings provide a screened set of epitopes that can help guide experimental efforts towards the development of vaccines against SARS-CoV-2.

## 1. Introduction

The ongoing outbreak of COVID-19 in the Chinese city of Wuhan (Hubei province) [1] and its alarmingly quick transmission to 25 other countries across the world [2] resulted in the World Health Organization (WHO) declaring a global health emergency on 30 January 2020 [3]. This came just one month after the first reported case on 31 December 2019 [4]. WHO, in its first emergency meeting [5], estimated the fatality rate of COVID-19 to be around 4%. Worldwide collaborative efforts from scientists are underway to understand the novel and rapidly spreading virus that causes this disease, SARS-CoV-2 (originally tentatively named 2019-nCoV), and to develop effective interventions for controlling and preventing it [6,7,8,9].

Coronaviruses are positive-sense single-stranded RNA viruses belonging to the family Coronaviridae. These viruses mostly infect animals, including birds and mammals. In humans, they generally cause mild respiratory infections, such as those observed in the common cold. However, some recent human coronavirus infections have resulted in lethal endemics, which include the SARS (Severe Acute Respiratory Syndrome) and MERS (Middle East Respiratory Syndrome) endemics. Both of these are caused by zoonotic coronaviruses that belong to the genus Betacoronavirus within Coronaviridae. SARS-CoV originated from Southern China and caused an endemic in 2003. A total of 8098 cases of SARS were reported globally, including 774 associated deaths, and an estimated case-fatality rate of 14%–15% [10]. The first case of MERS occurred in Saudi Arabia in 2012. Since then, a total of 2,494 cases of infection have been reported, including 858 associated deaths, and an estimated high case-fatality rate of 34.4% [11]. While no case of SARS-CoV infection has been reported since 2004, MERS-CoV has been around since 2012 and has caused multiple sporadic outbreaks in different countries. 

Like SARS-CoV and MERS-CoV, the recent SARS-CoV-2 belongs to the Betacoronavirus genus [12]. It has a genome size of ~30 kilobases which, like other coronaviruses, encodes for multiple structural and non-structural proteins. The structural proteins include the spike (S) protein, the envelope (E) protein, the membrane (M) protein, and the nucleocapsid (N) protein. With SARS-CoV-2 being discovered very recently, there is currently a lack of immunological information available about the virus (e.g., information about immunogenic epitopes eliciting antibody or T cell responses). Preliminary studies suggest that SARS-CoV-2 is quite similar to SARS-CoV based on the full-length genome phylogenetic analysis [9,12], and the putatively similar cell entry mechanism and human cell receptor usage [9,13,14]. Due to this apparent similarity between the two viruses, previous research that has provided an understanding of protective immune responses against SARS-CoV may potentially be leveraged to aid vaccine development for SARS-CoV-2. 

Various reports related to SARS-CoV suggest a protective role of both humoral and cell-mediated immune responses. For the former case, antibody responses generated against the S protein, the most exposed protein of SARS-CoV, have been shown to protect from infection in mouse models [15,16,17]. In addition, multiple studies have shown that antibodies generated against the N protein of SARS-CoV, a highly immunogenic and abundantly expressed protein during infection [18], were particularly prevalent in SARS-CoV-infected patients [19,20]. While being effective, the antibody response was found to be short-lived in convalescent SARS-CoV patients [21]. In contrast, T cell responses have been shown to provide long-term protection [21,22,23], even up to 11 years post-infection [24], and thus have also attracted interest for a prospective vaccine against SARS-CoV [reviewed in [25]]. Among all SARS-CoV proteins, T cell responses against the structural proteins have been found to be the most immunogenic in peripheral blood mononuclear cells of convalescent SARS-CoV patients as compared to the non-structural proteins [26]. Further, of the structural proteins, T cell responses against the S and N proteins have been reported to be the most dominant and long-lasting [27]. 

Here, by analyzing available experimentally-determined SARS-CoV-derived B cell epitopes (both linear and discontinuous) and T cell epitopes, we identify and report those that are completely identical and comprise no mutation in the available SARS-CoV-2 sequences (as of 21 February 2020). These epitopes have the potential, therefore, to elicit a cross-reactive/effective response against SARS-CoV-2. We focused particularly on the epitopes in the S and N structural proteins due to their dominant and long-lasting immune response previously reported against SARS-CoV. For the identified T cell epitopes, we additionally incorporated the information about the associated MHC alleles to provide a list of epitopes that seek to maximize population coverage globally, as well as in China. Our presented results can potentially narrow down the search for potent targets for an effective vaccine against SARS-CoV-2, and help guide experimental studies focused on vaccine development.

## 2. Materials and Methods

### 2.1. Acquisition and Processing of Sequence Data

A total of 120 whole genome sequences of SARS-CoV-2 were downloaded on 21 February 2020 from the GISAID database (https://www.gisaid.org/CoV2020/) (Appendix A). We excluded sequences that likely had spurious mutations resulting from sequencing errors, as indicated in the comment field of the GISAID data. These nucleotide sequences were aligned to the GenBank reference sequence (accession ID: NC_045512.2) and then translated into amino acid residues according to the coding sequence positions provided along the reference sequence for SARS-CoV-2 proteins (orf1a, orf1b, S, ORF3a, E, M, ORF6, ORF7a, ORF7b, ORF8, N, and ORF10). These sequences were aligned separately for each protein using the MAFFT multiple sequence alignment program [28]. Reference protein sequences for SARS-CoV and MERS-CoV were obtained following the same procedure from GenBank using the accession IDs NC_004718.3 and NC_019843.3, respectively.

### 2.2. Acquisition and Filtering of Epitope Data

SARS-CoV-derived B cell and T cell epitopes were searched on the NIAID Virus Pathogen Database and Analysis Resource (ViPR) (https://www.viprbrc.org/; accessed 21 February 2020) [29] by querying for the virus species name: “Severe acute respiratory syndrome-related coronavirus” from “human” hosts. We limited our search to include only the experimentally-determined epitopes that were associated with at least one positive assay: (i) Positive B cell assays (e.g., enzyme-linked immunosorbent assay (ELISA)-based qualitative binding) for B cell epitopes; and (ii) either positive T cell assays (such as enzyme-linked immune absorbent spot (ELISPOT) or intracellular cytokine staining (ICS) IFN-γ release), or positive major histocompatibility complex (MHC) binding assays for T cell epitopes. Strictly speaking, the latter set of epitopes, determined using positive MHC binding assays, are antigens which are candidate epitopes, since a T cell response has not been confirmed experimentally. However, for brevity and to be consistent with the terminology used in the ViPR database, we will not make this qualification, and will simply refer to them as epitopes in this study. The number of B cell and T cell epitopes obtained from the database following the above procedure is listed in Table 1.

### 2.3. Population-Coverage-Based T Cell Epitope Selection

Population coverages for sets of T cell epitopes were computed using the tool provided by the Immune Epitope Database (IEDB) (http://tools.iedb.org/population/; accessed 21 February 2020) [30]. This tool uses the distribution of MHC alleles (with at least 4-digit resolution, e.g., A*02:01) within a defined population (obtained from http://www.allelefrequencies.net/) to estimate the population coverage for a set of T cell epitopes. The estimated population coverage represents the percentage of individuals within the population that are likely to elicit an immune response to at least one T cell epitope from the set. To identify the set of epitopes associated with MHC alleles that would maximize the population coverage, we adopted a greedy approach: (i) We first identified the MHC allele with the highest individual population coverage and initialized the set with their associated epitopes, then (ii) we progressively added epitopes associated with other MHC alleles that resulted in the largest increase of the accumulated population coverage. We stopped when no increase in the accumulated population coverage was observed by adding epitopes associated with any of the remaining MHC alleles.

### 2.4. Constructing the Phylogenetic Tree

We used the publicly available software PASTA v1.6.4 [31] to construct a maximum-likelihood phylogenetic tree of each structural protein using the unique set of sequences in the available data of SARS-CoV, MERS-CoV, and SARS-CoV-2. We additionally included the Zaria Bat coronavirus strain (accession ID: HQ166910.1) to serve as an outgroup. The appropriate parameters for tree estimation are automatically selected in the software based on the provided sequence data. For visualizing the constructed phylogenetic trees, we used the publicly available software Dendroscope v3.6.3 [32]. Each constructed tree was rooted with the outgroup Zaria Bat coronavirus strain, and circular phylogram layout was used.

### 2.5. Data and Code Availability

All sequence and immunological data, and all scripts (written in R) for reproducing the results are available online [33].

## 3. Results

### 3.1. Structural Proteins of SARS-CoV-2 Are Genetically Similar to SARS-CoV, but Not to MERS-CoV

SARS-CoV-2 has been observed to be close to SARS-CoV—much more so than MERS-CoV—based on full-length genome phylogenetic analysis [9,12]. We checked whether this is also true at the level of the individual structural proteins (S, E, M, and N). A straightforward reference-sequence-based comparison indeed confirmed this, showing that the M, N, and E proteins of SARS-CoV-2 and SARS-CoV have over 90% genetic similarity, while that of the S protein was notably reduced (but still high) (Figure 1a). The similarity between SARS-CoV-2 and MERS-CoV, on the other hand, was substantially lower for all proteins (Figure 1a); a feature that was also evident from the corresponding phylogenetic trees (Figure 1b). We note that while the former analysis (Figure 1a) was based on the reference sequence of each coronavirus, it is indeed a good representative of the virus population, since few amino acid mutations have been observed in the corresponding sequence data (Appendix A). It is also noteworthy that while MERS-CoV is the more recent coronavirus to have infected humans, and is comparatively more recurrent (causing outbreaks in 2012, 2015, and 2018) (https://www.who.int/emergencies/mers-cov/en/), SARS-CoV-2 is closer to SARS-CoV, which has not been observed since 2004. 

Given the close genetic similarity between the structural proteins of SARS-CoV and SARS-CoV-2, we attempted to leverage immunological studies of the structural proteins of SARS-CoV to potentially aid vaccine development for SARS-CoV-2. We focused specifically on the S and N proteins as these are known to induce potent and long-lived immune responses in SARS-CoV [15,16,17,19,20,25,27]. We used the available SARS-CoV-derived experimentally-determined epitope data (see Materials and Methods) and searched to identify T cell and B cell epitopes that were identical—and hence potentially cross-reactive—across SARS-CoV and SARS-CoV-2. We first report the analysis for T cell epitopes, which have been shown to provide a long-lasting immune response against SARS-CoV [27], followed by a discussion of B cell epitopes.

### 3.2. Mapping the SARS-CoV-Derived T Cell Epitopes That Are Identical in SARS-CoV-2, and Determining Those With Greatest Estimated Population Coverage

The SARS-CoV-derived T cell epitopes used in this study were experimentally-determined from two different types of assays [29]: (i) Positive T cell assays, which tested for a T cell response against epitopes, and (ii) positive MHC binding assays, which tested for epitope-MHC binding. We aligned these T cell epitopes across the SARS-CoV-2 protein sequences. Among the 115 T cell epitopes that were determined by positive T cell assays (Table 1), we found that 27 epitope-sequences were identical within SARS-CoV-2 proteins and comprised no mutation in the available SARS-CoV-2 sequences (as of 21 February 2020) (Table 2). Interestingly, all of these were present in either the N (16) or S (11) protein. MHC binding assays were performed for 19 of these 27 epitopes, and these were reported to be associated with only five distinct MHC alleles (at 4-digit resolution): HLA-A*02:01, HLA-B*40:01, HLA-DRA*01:01, HLA-DRB1*07:01, and HLA-DRB1*04:01. Consequently, the accumulated population coverage of these epitopes (see Materials and Methods for details) is estimated to not be high for the global population (59.76%), and was quite low for China (32.36%). For the remaining 8 epitopes, since the associated MHC alleles are unknown, they could not be used in the population coverage computation. Additional MHC binding tests to identify the MHC alleles that bind to these 8 epitopes may reveal additional distinct alleles, beyond the five determined so far, that may help to improve population coverage.

To further expand the search and identify potentially effective T cell targets covering a higher percentage of the population, we next additionally considered the set of T cell epitopes that have been experimentally-determined from positive MHC binding assays (Table 1), but, unlike the previous epitope set, their ability to induce a T cell response against SARS-CoV was not experimentally determined. Nonetheless, they also present promising candidates for inducing a response against SARS-CoV-2. For the expanded set of epitopes, all of which have at least one positive MHC binding assay, we found that 229 epitope-sequences have an identical match in SARS-CoV-2 proteins and have associated MHC allele information available (listed in Appendix A). Of these 229 epitopes, ~82% were MHC Class I restricted epitopes (Appendix A). Importantly, 102 of the 229 epitopes were derived from either the S (66) or N (36) protein. Mapping all 66 S-derived epitopes onto the resolved crystal structure of the SARS-CoV S protein (Appendix A) revealed that 3 of these (GYQPYRVVVL, QPYRVVVLSF, and PYRVVVLSF) were located entirely in the SARS-CoV receptor-binding motif (https://www.uniprot.org/uniprot/P59594), known to be important for virus cell entry [34]. 

Similar to previous studies on HIV and HCV [35,36,37,38], we estimated population coverages for various combinations of MHC alleles associated with these 102 epitopes. Our aim was to determine sets of epitopes associated with MHC alleles with maximum population coverage, potentially aiding the development of vaccines against SARS-CoV-2. For selection, we adopted a greedy computational approach (see Materials and Methods), which identified a set of T cell epitopes estimated to maximize global population coverage. This set comprised of multiple T cell epitopes associated with 20 distinct MHC alleles and was estimated to provide an accumulated population coverage of 96.29% (Table 3). Interestingly, the majority of the T cell epitopes for which a positive immune response has been determined using T cell assays (Table 2) were presented by the globally most-prevalent MHC allele (shown in blue color in Table 3). Moreover, the functionally important epitopes located in the SARS-CoV receptor binding motif were associated with the second and third most-prevalent MHC alleles (underlined in Table 3). Thus, while the ordering of T cell epitopes in Table 3 is based on the estimated global population coverage of the associated MHC alleles, it is also a natural order in which these epitopes should be tested experimentally for determining their potential to induce a positive immune response against SARS-CoV-2. We also computed the population coverage of this specific set of epitopes in China, the country most affected by the COVID-19 outbreak, which was estimated to be slightly lower (88.11%), as certain MHC alleles (e.g., HLA-A*02:01) associated with some of these epitopes are less frequent in the Chinese population (Table 3). Repeating the same greedy approach but focusing on the Chinese population, instead of a global population, the maximum population coverage was estimated to be 92.76% (Appendix A).

Due to the promiscuous nature of binding between peptides and MHC alleles, multiple S and N peptides were reported to bind to individual MHC alleles. Thus, while we list all the S and N epitopes that bind to each MHC allele (Table 3), the estimated maximum population coverage may be achieved by selecting at least one epitope for each listed MHC allele. Likewise, many individual S and N epitopes were found to be presented by multiple alleles and thereby estimated to have varying global population coverage (listed in Appendix A).

### 3.3. Mapping the SARS-CoV-Derived B cell Epitopes that Are Identical in SARS-CoV-2

Similar to T cell epitopes, we used in our study the SARS-CoV-derived B cell epitopes that have been experimentally-determined from positive B cell assays [29]. These epitopes were classified as: (i) Linear B cell epitopes (antigenic peptides), and (ii) discontinuous B cell epitopes (conformational epitopes with resolved structural determinants).

We aligned the 298 linear B cell epitopes (Table 1) across the SARS-CoV-2 proteins and found that 49 epitope-sequences, all derived from structural proteins, have an identical match and comprised no mutation in the available SARS-CoV-2 protein sequences (as of 21 February 2020). Interestingly, a large number (45) of these were derived from either the S (23) or N (22) protein (Table 4), while the remaining (4) were from the M protein (Appendix A).

On the other hand, all 6 SARS-CoV-derived discontinuous B cell epitopes obtained from the ViPR database (Table 5) were derived from the S protein. Based on the pairwise alignment between the SARS-CoV and SARS-CoV-2 reference sequences (Appendix A), we found that none of these mapped identically to the SARS-CoV-2 S protein, in contrast to the linear epitopes. For 3 of these discontinuous B cell epitopes (corresponding to antibodies S230, m396, and 80R [39,40,41]), there was a partial mapping, with at least one site having an identical residue at the corresponding site in the SARS-CoV-2 S protein (Table 5). 

Mapping the residues of the linear and discontinuous B cell epitopes onto the available structure of the SARS-CoV S protein revealed their distinct association with the two functional subunits of the S protein [42]: S1, important for interaction with the host cell receptor, and S2, involved in fusion of the cellular and virus membranes (Figure 2a). Specifically, 20 of the 23 linear epitopes (Table 4) mapped to S2 (Figure 2b). Thus, the antibodies targeting the identified linear epitopes in the S2 subunit might cross-react and neutralize both SARS-CoV and SARS-CoV-2, as suggested in a very recent study [43]. While S2 is comparatively less exposed than S1, it may be accessible to antibodies during the complex conformational changes involved in viral entry of coronaviruses [44,45,46]; though this remains to be more clearly understood. In contrast, the 3 discontinuous B cell epitopes (Table 5) mapped onto the more exposed S1 subunit (Figure 2c, left panel), which contains the receptor-binding motif of the SARS-CoV S protein [34]. We observed that very few residues of the 3 discontinuous epitopes were identical within SARS-CoV and SARS-CoV-2 (Figure 2c, right panel). These differences suggest that the SARS-CoV-specific antibodies S230, m396, and 80R known to bind to these epitopes in SARS-CoV might not be able to bind to the same regions in SARS-CoV-2 S protein. Interestingly, while this paper was under review, this has been confirmed experimentally [47]. Further studies are currently under way to identify other SARS-CoV antibodies that may bind to discontinuous epitopes of the SARS-CoV-2 S protein [48]. 

## 4. Discussion

The quest for a vaccine against the novel SARS-CoV-2 is recognized as an urgent problem. Effective vaccination could indeed play a significant role in curbing the spread of the virus, and help to eliminate it from the human population. However, scientific efforts to address this challenge are only just beginning. Much remains to be learnt about the virus, its biological properties, epidemiology, etc. At this early stage, there is also a lack of information about specific immune responses against SARS-CoV-2, which presents a challenge for vaccine development. 

This study has sought to assist with the initial phase of vaccine development by providing recommendations of epitopes that may potentially be considered for incorporation in vaccine designs. Despite having limited understanding of how the human immune system responds naturally to SARS-CoV-2, these epitopes are motivated by responses they have recorded in SARS-CoV (or, for the case of T cell epitopes, to at least confer MHC binding), and the fact that they map identically to SARS-CoV-2, based on the available sequence data (as of 21 February 2020). This important observation should not be taken for granted. Despite the apparent similarity between SARS-CoV and SARS-CoV-2, there is still considerable genetic variation between the two, and it is not obvious a-prior if epitopes that elicit an immune response against SARS-CoV are likely to be effective against SARS-CoV-2. We found that only 23% and 16% of known SARS-CoV T cell and B cell epitopes map identically to SARS-CoV-2, respectively, and with no mutation having been observed in these epitopes among the available SARS-CoV-2 sequences (as of 21 February 2020). This provides a strong indication of their potential for eliciting a robust T cell or antibody response in SARS-CoV-2.

On the T cell side, the identification of SARS-CoV-derived epitopes that map identically to SARS-CoV-2, and the large population that these are expected to cover, is particularly encouraging. It promotes further research in exploring vaccines designed to induce a protective T cell response, which has been shown to provide long term protection in SARS-CoV [21,22,23,24]. On the B cell side, in agreement with very recent experimental studies [47,48], our results suggest that SARS-CoV-derived antibodies targeting the receptor binding motif in the S1 subunit of the SARS-CoV-2 S protein may not be effective, due to the large genetic mismatches observed in known structural epitopes targeting this domain. Linear SARS-CoV-derived B cell epitopes in the S2 subunit may potentially be more promising candidates for inducing a protective antibody response. Numerous of these epitopes, while being less exposed, are found to map identically to SARS-CoV and SARS-CoV-2, and preliminary results are already emerging which suggest their potential in generating cross-reactive and neutralizing antibodies [43]. Hence, vaccine solutions that attempt to induce antibodies that target the S2 linear epitopes may be effective and should be explored further.

Research efforts directed towards the design and development of vaccines for SARS-CoV-2 are increasing, and some related analyses are already being reported in distinct, parallel studies. These studies, like our own, are based on leveraging available data and computational methods, and add to recent work focused on computational analysis and design of vaccines for various different viruses (e.g., [49,50,51,52,53,54]. A preliminary analysis of linear SARS-CoV-derived B cell epitopes has been reported online on the ViPR database website (https://www.viprbrc.org/brcDocs/documents/announcements/Corona/2019-nCoV-ViPR-report_24JAN2020.pdf). Different from our study, which is focused on the linear and discontinuous SARS-CoV-derived epitopes, that analysis considered linear B cell epitope data for all Betacoronaviruses from human hosts. While only a summary of the results has been provided so far, preventing direct comparison of the individual epitopes, the number of linear B cell epitopes reported to map identically to SARS-CoV-2 is comparable to our findings. 

A recent study has also predicted T cell epitopes for SARS-CoV-2 that may be presented by a population from the Asia-Pacific region [55]. Again, there are multiple differences to our work. First, the focus of that study was on MHC Class II epitopes, while here we considered both MHC Class I and II epitopes. Interestingly, while we found a few MHC Class II epitopes using our approach (Appendix A), only one of these (HLA-DRB1*01:01) appeared in our identified epitope set (Table 3), due to their comparatively low estimated population coverage. Second, computational tools were used to predict MHC Class II epitopes in [55], while here we analyzed the SARS-CoV-derived epitopes that have been determined experimentally, using either positive T cell or MHC binding assays, and which match identically with the available SARS-CoV-2 sequences (as of 21 February 2020). Thus, our identified epitopes are seemingly a more rational set of potential targets that can assist in the ongoing search for a SARS-CoV-2 vaccine.

We acknowledge that this is a preliminary analysis based on the limited sequence data available for SARS-CoV-2 (as of 21 February 2020). As the virus continues to evolve and as more data is collected, it is expected that additional mutations will be observed. Such mutations will not affect our analysis, provided that they occur outside of the identified epitope regions. If mutations do occur within epitope regions, then these epitopes may be further screened in line with the conservative filtering principle that we have employed, thereby producing a more refined epitope set.

Further experimental studies (T cell and B cell assays) are required to determine the potential of the identified epitopes to induce a positive immune response against SARS-CoV-2. This would help to further refine the reported epitope set, based on observed immunogenicity; an important consideration for immunogen design. 

Overall, as the identified set of SARS-CoV epitopes map identically to SARS-CoV-2, they present potentially useful candidates for guiding experimental efforts towards developing vaccines against SARS-CoV-2. More generally, our study further highlights the potential importance of previous experimental and clinical studies of SARS-CoV, and its use in concert with emerging data for SARS-CoV-2, in searching for effective vaccines to combat the COVID-19 epidemic. 

## Figures and Tables

**Figure 1 viruses-12-00254-f001:**
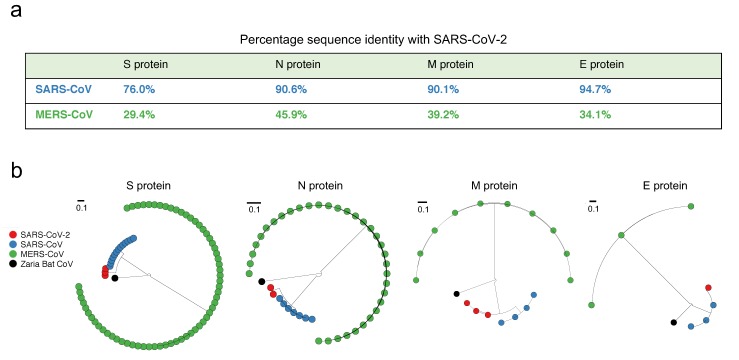
Comparison of the similarity of structural proteins of SARS-CoV-2 with the corresponding proteins of SARS-CoV and MERS (Middle East Respiratory Syndrome)-CoV. (**a**) Percentage genetic similarity of the individual structural proteins of SARS-CoV-2 with those of SARS-CoV and MERS-CoV. The reference sequence of each coronavirus (Materials and Methods) was used to calculate the percentage genetic similarity. (**b**) Circular phylogram of the phylogenetic trees of the four structural proteins. All trees were constructed based on the available unique sequences using PASTA [31] and rooted with the outgroup Zaria Bat CoV strain (accession ID: HQ166910.1).

**Figure 2 viruses-12-00254-f002:**
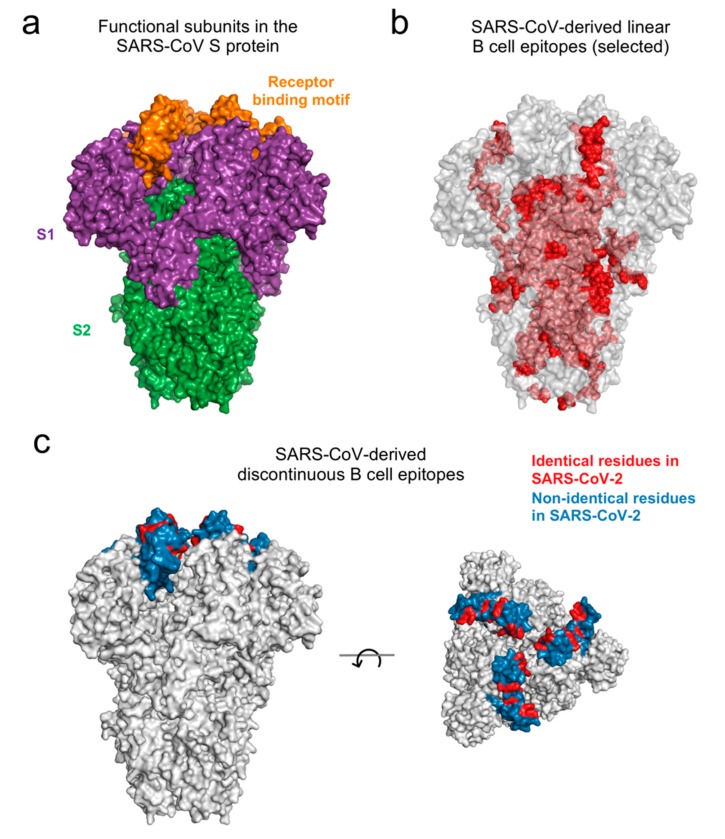
Location of SARS-CoV S protein subunits and SARS-CoV-derived B cell epitopes on the protein structure (PDB ID: 5XLR). (**a**) Subunits S1 and S2 are indicated in purple and green color, respectively. The receptor binding motif lies within the S1 subunit and is indicated in orange color. (**b**) Residues of the linear B cell epitopes, that were identical in SARS-CoV-2 (Table 4), are shown in red color. The dark and light shade reflect the surface and buried residues, respectively. (**c**) Location of discontinuous B cell epitopes that share at least one identical residue with corresponding SARS-CoV-2 sites (Table 5). Identical epitope residues are shown in red color, while the remaining epitope residues are shown in blue color. Both the side view (*left panel*) and the top view (*right panel*) of the structure are shown.

**Table 1 viruses-12-00254-t001:** Filtering criteria and corresponding number of Severe Acute Respiratory Syndrome Coronavirus (SARS-CoV)-derived epitopes obtained from the Virus Pathogen Database and Analysis Resource (ViPR) database.

Filtering Criteria		Number of Epitopes
Positive T cell assays	T cell epitopes	115
Positive major histocompatibility complex (MHC) binding assays	T cell epitopes	959
Positive B cell assays	Linear B cell epitopes	298
	Discontinuous B cell epitopes	6

**Table 2 viruses-12-00254-t002:** SARS-CoV-derived T cell epitopes obtained using positive T cell assays that are identical in SARS-CoV-2 (27 epitopes in total).

Protein	IEDB ID	Epitope	MHC Allele^1^	MHC Allele Class ^1^
N	125100	ILLNKHID	HLA-A*02:01	I
N	1295	AFFGMSRIGMEVTPSGTW	NA	NA
N	190494	MEVTPSGTWL	HLA-B*40:01	I
N	21347	GMSRIGMEV	HLA-A*02:01	I
N	27182	ILLNKHIDA	HLA-A*02:01	I
N	2802	ALNTPKDHI	HLA-A*02:01	I
N	28371	IRQGTDYKHWPQIAQFA	NA	NA
N	31166	KHWPQIAQFAPSASAFF	NA	NA
N	34851	LALLLLDRL	HLA-A*02:01	I
N	37473	LLLDRLNQL	HLA-A*02:01	I
N	37611	LLNKHIDAYKTFPPTEPK	NA	NA
N	38881	LQLPQGTTL	HLA-A*02:01	I
N	3957	AQFAPSASAFFGMSR	NA	II
N	3958	AQFAPSASAFFGMSRIGM	NA	NA
N	55683	RRPQGLPNNTASWFT	NA	I
N	74517	YKTFPPTEPKKDKKKK	NA	NA
S	100048	GAALQIPFAMQMAYRF	HLA-DRA*01:01, HLA-DRB1*07:01	II
S	100300	MAYRFNGIGVTQNVLY	HLA-DRB1*04:01	II
S	100428	QLIRAAEIRASANLAATK	HLA-DRB1*04:01	II
S	16156	FIAGLIAIV	HLA-A*02:01	I
S	2801	ALNTLVKQL	HLA-A*02:01	I
S	36724	LITGRLQSL	HLA-A2	I
S	44814	NLNESLIDL	HLA-A*02:01	I
S	50311	QALNTLVKQLSSNFGAI	HLA-DRB1*04:01	II
S	54680	RLNEVAKNL	HLA-A*02:01	I
S	69657	VLNDILSRL	HLA-A*02:01	I
S	71663	VVFLHVTYV	HLA-A*02:01	I

^1^ NA: Not available.

**Table 3 viruses-12-00254-t003:** Set of the SARS-CoV-derived spike (S) and nucleocapsid (N) protein T cell epitopes (obtained from positive MHC binding assays) that are identical in SARS-CoV-2 and that maximize estimated population coverage globally (87 distinct epitopes).

Epitopes^1^	MHC Allele Class	MHC Allele	Global Accumulated Population Coverage^2^ (%)	Accumulated Population Coverage in China (%)
**FIAGLIAIV**, GLIAIVMVTI, IITTDNTFV, **ALNTLVKQL**, LITGRLQSL, LLLQYGSFC, LQYGSFCT, **NLNESLIDL**, RLDKVEAEV, **RLNEVAKNL**, RLQSLQTYV, **VLNDILSRL**, **VVFLHVTYV**, **ILLNKHID**, FPRGQGVPI, LLLLDRLNQ, **GMSRIGMEV**, **ILLNKHIDA**, **ALNTPKDHI**, **LALLLLDRL**, **LLLDRLNQL**, LLLLDRLNQL, **LQLPQGTTL**, AQFAPSASA, TTLPKGFYA, VLQLPQGTTL	I	HLA-A*02:01	39.08	14.62
GYQPYRVVVL, PYRVVVLSF, LSPRWYFYY	I	HLA-A*24:02	55.48	36.11
DSFKEELDKY, LIDLQELGKY, PYRVVVLSF, GTTLPKGFY, VTPSGTWLTY	I	HLA-A*01:01	66.78	39.09
GSFCTQLNR, GVVFLHVTY, AQALNTLVK, MTSCCSCLK, ASANLAATK, SLIDLQELGK, SVLNDILSR, TQNVLYENQK, CMTSCCSCLK, VQIDRLITGR, KTFPPTEPK, KTFPPTEPKK, LSPRWYFYY, ASAFFGMSR, ATEGALNTPK, QLPQGTTLPK, QQQGQTVTK, QQQQGQTVTK, SASAFFGMSR, SQASSRSSSR, TPSGTWLTY	I	HLA-A*03:01	76.14	41.68
GSFCTQLNR, GVVFLHVTY, AQALNTLVK, MTSCCSCLK, ASANLAATK, SLIDLQELGK, SVLNDILSR, TQNVLYENQK, CMTSCCSCLK, VQIDRLITGR, KTFPPTEPK, KTFPPTEPKK, LSPRWYFYY, ASAFFGMSR, ATEGALNTPK, QLPQGTTLPK, QQQGQTVTK, QQQQGQTVTK, SASAFFGMSR, SQASSRSSSR, TPSGTWLTY	I	HLA-A*11:01	83.39	73.43
GSFCTQLNR, GVVFLHVTY, AQALNTLVK, MTSCCSCLK, ASANLAATK, SLIDLQELGK, SVLNDILSR, TQNVLYENQK, CMTSCCSCLK, VQIDRLITGR, KTFPPTEPK, KTFPPTEPKK, LSPRWYFYY, ASAFFGMSR, ATEGALNTPK, QLPQGTTLPK, QQQGQTVTK, QQQQGQTVTK, SASAFFGMSR, SQASSRSSSR, TPSGTWLTY	I	HLA-A*68:01	85.71	74.25
GYQPYRVVVL, PYRVVVLSF, LSPRWYFYY	I	HLA-A*23:01	87.72	74.87
GSFCTQLNR, GVVFLHVTY, AQALNTLVK, MTSCCSCLK, ASANLAATK, SLIDLQELGK, SVLNDILSR, TQNVLYENQK, CMTSCCSCLK, VQIDRLITGR, KTFPPTEPK, KTFPPTEPKK, LSPRWYFYY, ASAFFGMSR, ATEGALNTPK, QLPQGTTLPK, QQQGQTVTK, QQQQGQTVTK, SASAFFGMSR, SQASSRSSSR, TPSGTWLTY	I	HLA-A*31:01	89.55	76.93
FPNITNLCPF, APHGVVFLHV, FPRGQGVPI, APSASAFFGM	I	HLA-B*07:02	90.89	77.61
GAALQIPFAMQMAYR, GWTFGAGAALQIPFA, IDRLITGRLQSLQTY, ISGINASVVNIQKEI, LDKYFKNHTSPDVDL, LGDISGINASVVNIQ, LGFIAGLIAIVMVTI, LNTLVKQLSSNFGAI, LQDVVNQNAQALNTL, LQSLQTYVTQQLIRA, LQTYVTQQLIRAAEI, AQKFNGLTVLPPLLT, PCSFGGVSVITPGTN, QIPFAMQMAYRFNGI, QQLIRAAEIRASANL, QTYVTQQLIRAAEIR, AYRFNGIGVTQNVLY, SSNFGAISSVLNDIL, TGRLQSLQTYVTQQL, WLGFIAGLIAIVMVT, CVNFNFNGLTGTGVL, DKYFKNHTSPDVDLG, IDAYKTFPPTEPKKD, MSRIGMEVTPSGTWL, NKHIDAYKTFPPTEP, VLQLPQGTTLPKGFY	II	HLA-DRB1*01:01	91.94	78.23
FPRGQGVPI	I	HLA-B*08:01	92.85	78.41
FPNITNLCPF, APHGVVFLHV, FPRGQGVPI, APSASAFFGM	I	HLA-B*35:01	93.53	79.23
LQIPFAMQM, RVDFCGKGY	I	HLA-B*15:01	94.18	82.26
FPNITNLCPF, APHGVVFLHV, FPRGQGVPI, APSASAFFGM	I	HLA-B*51:01	94.72	83.73
YEQYIKWPWY	I	HLA-B*18:01	95.23	83.88
GRLQSLQTY, RVDFCGKGY, VRFPNITNL	I	HLA-B*27:05	95.55	84
MTSCCSCLK, SLIDLQELGK, CMTSCCSCLK, VQIDRLITGR, SASAFFGMSR, SQASSRSSSR	I	HLA-A*33:01	95.79	85.28
LQIPFAMQM, RVDFCGKGY	I	HLA-B*58:01	95.99	86.45
LQIPFAMQM, RVDFCGKGY	I	HLA-C*15:02	96.17	87.22
VRFPNITNL	I	HLA-C*14:02	96.29	88.11

^1^ Multiple SARS-CoV-derived epitopes that were determined using MHC binding assays are shown for each allele. Epitopes that were also tested for positive T cell response (listed also in Table 2) are shown in bold. Epitopes that lie within the SARS-CoV receptor-binding motif are underlined. ^2^ Epitopes are ordered according to the estimated global accumulated population coverage.

**Table 4 viruses-12-00254-t004:** SARS-CoV-derived linear B cell epitopes from S (23; 20 of which are located in subunit S2) and N (22) proteins that are identical in SARS-CoV-2 (45 epitopes in total).

Protein	Subunit	IEDB ID	Epitope	Protein	IEDB ID	Epitope
S	S2	10778	DVVNQNAQALNTLVKQL	N	15814	FFGMSRIGMEVTPSGTW
S	S2	11038	EAEVQIDRLITGRLQSL	N	21065	GLPNNTASWFTALTQHGK
S	S2	12426	EIDRLNEVAKNLNESLIDLQELGKYEQY	N	22855	GTTLPK
S	S2	14626	EVAKNLNESLIDLQELG	N	28371	IRQGTDYKHWPQIAQFA
S	S2	18515	GAALQIPFAMQMAYRFN	N	31116	KHIDAYKTFPPTEPKKDKKK
S	S1	18594	GAGICASY	N	31166	KHWPQIAQFAPSASAFF
S	S2	2092	AISSVLNDILSRLDKVE	N	75235	YNVTQAFGRRGPEQTQGNF
S	S2	22321	GSFCTQLN	N	33669	KTFPPTEPKKDKKKK
S	S2	27357	ILSRLDKVEAEVQIDRL	N	37640	LLPAAD
S	S1	30987	KGIYQTSN	N	38249	LNKHIDAYKTFPPTEPK
S	S2	3176	AMQMAYRF	N	38648	LPQGTTLPKG
S	S2	32508	KNHTSPDVDLGDISGIN	N	38657	LPQRQKKQ
S	S2	41177	MAYRFNGIGVTQNVLYE	N	48067	PKGFYAEGSRGGSQASSR
S	S2	462	AATKMSECVLGQSKRVD	N	50741	QFAPSASAFFGMSRIGM
S	S2	47479	PFAMQMAYRFNGIGVTQ	N	50965	QGTDYKHW
S	S2	50311	QALNTLVKQLSSNFGAI	N	51483	QLPQGTTLPKGFYAE
S	S2	51379	QLIRAAEIRASANLAAT	N	51484	QLPQGTTLPKGFYAEGSR
S	S1	52020	QQFGRD	N	51485	QLPQGTTLPKGFYAEGSRGGSQ
S	S2	53202	RASANLAATKMSECVLG	N	63729	TFPPTEPK
S	S2	54599	RLITGRLQSLQTYVTQQ	N	55683	RRPQGLPNNTASWFT
S	S2	558417	EIDRLNEVAKNLNESLIDLQELGKYEQY	N	60379	SQASSRSS
S	S2	59425	SLQTYVTQQLIRAAEIR	N	60669	SRGGSQASSRSSSRSR
S	S2	9094	DLGDISGINASVVNIQK			

**Table 5 viruses-12-00254-t005:** SARS-CoV-derived discontinuous B cell epitopes (and associated known antibodies [39,40,41]) that have at least one site with an identical amino acid to the corresponding site in SARS-CoV-2.

IEDB ID	Associated Known Antibody	SARS-CoV S Protein Residues ^1,2^
910052	S230	G446, P462, D463, Y475
77444	m396	T359, T363, K365, K390, G391, D392, R395, R426, Y436, G482, Y484, T485, T486, T487, G488, I489, G490, Y491, Q492, Y494
77442	80R	R426, S432, T433, Y436, N437, K439, Y440, Y442, P469, P470, A471, L472, N473, C474, Y475, W476, L478, N479, D480, Y481, G482, Y484, T485, T486, T487, G488, I489, Y491, Q492

^1^ Residues are numbered according to the SARS-CoV S protein reference sequence, accession ID: NP_828851.1.; ^2^ Residues in the epitopes that are identical in the SARS-CoV-2 sequences are underlined.

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
