# Peer review of "Preliminary Identification of Potential Vaccine Targets for the COVID-19 Coronavirus (SARS-CoV-2) Based on SARS-CoV Immunological Studies"

_viruses, 2020, doi:10.3390/v12030254_

Round 1
Reviewer 1 Report
The merit of this submission is with its bioinformatic analysis of potential SARS-2 CoV B and T cell epitopes. Similar bioinformatic analyses have been presented, and therefore the novelty of the report is somewhat low. There are no experiments performed and therefore the biological significance of the bioinformatic results may be questioned. In this regard, the B cell epitopes of relevance are in the spike RBD, and not in the interior regions of the spike (this should be noted and indicated more clearly in one of the figures). The RBD is the principal site at which bound antibodies neutralize betacoronaviruses. Yet, very recent reports in biorvix, for example, https://doi.org/10.1101/2020.02.11.944462, indicate that known SARS-CoV RBD neutralizing antibodies do not bind to SARS-2 CoV. This raises questions about whether the bioinformatic prediction of SARS-2 CoV B cell epitopes is credible and whether the predictive epitopes will turn out to be useful in vaccine design. Predictive algorithms for T cell epitopes may have more utility but even here there is the need for experimental validations.
Reviewer 2 Report
A very timely paper describing the identification of a set of B cell and T cell epitopes on SARS-CoV proteins that map identically to SARS-CoV2 proteins. The idea is interesting. But I don't think the author used a method.
For T cell epitope, the author mentioned "a T cell response has not been confirmed experimentally". So I strongly doubt whether these data are solid and useful.
For B cell epitope, S protein is the only one on the viral surface and can induce human immune responce. It makes no sense to predict B cell epitope for other structural proteins. I don't know what's the meaning of these "B cell epitope".
